# Addressing Post-Acute COVID-19 Syndrome in Cancer Patients, from Visceral Obesity and Myosteatosis to Systemic Inflammation: Implications in Cardio-Onco-Metabolism

**DOI:** 10.3390/biomedicines12081650

**Published:** 2024-07-24

**Authors:** Vincenzo Quagliariello, Maria Laura Canale, Irma Bisceglia, Carlo Maurea, Domenico Gabrielli, Luigi Tarantini, Andrea Paccone, Alessandro Inno, Stefano Oliva, Christian Cadeddu Dessalvi, Concetta Zito, Michele Caraglia, Massimiliano Berretta, Giuseppe D’Aiuto, Nicola Maurea

**Affiliations:** 1Division of Cardiology, Istituto Nazionale Tumori-IRCCS-Fondazione G. Pascale, 80131 Napoli, Italyn.maurea@istitutotumori.na.it (N.M.); 2U.O.C. Cardiologia, Ospedale Versilia, 55041 Lido di Camaiore, Italy; marialaura.canale@uslnordovest.toscana.it; 3Servizi Cardiologici Integrati, Dipartimento Cardio-Toraco-Vascolare, Azienda Ospedaliera San Camillo Forlanini, 00152 Roma, Italy; irmabisceglia@gmail.com; 4Neurology Department, University of Salerno, 84084 Fisciano, Italy; carlo.maurea@libero.it; 5U.O.C. Cardiologia, Dipartimento Cardio-Toraco-Vascolare, Azienda Ospedaliera San Camillo Forlanini, 00152 Roma, Italy; dgabrielli@scamilloforlanini.rm.it; 6Divisione di Cardiologia, Arcispedale S. Maria Nuova, Azienda Unità Sanitaria Locale-IRCCS di Reggio-Emilia, 42122 Reggio Emilia, Italy; luigi.tarantini@ausl.re.it; 7Medical Oncology, IRCCS Ospedale Sacro Cuore Don Calabria, 37024 Negrar di Valpolicella, Italy; alessandro.inno@sacrocuore.it; 8UOSD Cardiologia di Interesse Oncologico IRCCS Istituto Tumori “Giovanni Paolo II”, 70124 Bari, Italy; s.oliva@oncologico.bari.it; 9Department of Medical Sciences and Public Health, University of Cagliari, 09042 Monserrato, Italy; cadedduc@unica.it; 10Cardiology Division, University Hospital Polyclinic G. Martino, University of Messina, 98122 Messina, Italy; concetta.zito@unime.it; 11Department of Precision Medicine, University of Campania “Luigi Vanvitelli”, Via L. De Crecchio, 7, 80138 Naples, Italy; michele.caraglia@unicampania.it; 12Department of Clinical and Experimental Medicine, University of Messina, 98122 Messina, Italy; berrettama@gmail.com; 13Clinica C.G. Ruesch, 80122 Naples, Italy; giuseppe@daiuto.org

**Keywords:** cancer, cardio-oncology, COVID-19, cardiotoxicity, inflammation, metabolism

## Abstract

Cardiovascular disease and cancer are the two leading causes of morbidity and mortality in the world. The emerging field of cardio-oncology described several shared risk factors that predispose patients to both cardiovascular disease and cancer. Post-acute COVID-19 syndrome is a chronic condition that occurs in many patients who have experienced a SARS-CoV-2 infection, mainly based on chronic fatigue, sedentary lifestyle, cramps, breathing difficulties, and reduced lung performance. Post-acute COVID-19 exposes patients to increased visceral adiposity, insulin resistance, myosteatosis, and white adipose tissue content (surrounded by M1 macrophages and characterized by a Th1/Th17 phenotype), which increases the risk of cardiovascular mortality and cancer recurrence. In this review, the main metabolic affections of post-acute COVID-19 syndrome in cancer patients at low and high risk of cardiomyopathies will be summarized. Furthermore, several non-pharmacological strategies aimed at reducing atherosclerotic and cardiac risk will be provided, especially through anti-inflammatory nutrition with a low insulin and glycemic index, appropriate physical activity, and immune-modulating bioactivities able to reduce visceral obesity and myosteatosis, improving insulin-related signaling and myocardial metabolism.

## 1. Introduction

Post-acute COVID-19 syndrome (PACS), also known as long COVID or long-haul COVID, refers to a multifactorial condition where patients experience persistent symptoms and complications following the acute phase of SARS-CoV-2 infection [1,2]. While most people recover from COVID-19 within a few weeks, some individuals, including cancer patients, may continue to experience symptoms for a prolonged period, exposing them to high cardiovascular and metabolic risks [3,4]. The specific symptoms experienced by cancer patients with PACS can vary widely and may include fatigue, shortness of breath, chest pain, cognitive difficulties (often referred to as “brain fog”), muscle or joint pain, sleep disturbances, and mood changes [5,6]. These symptoms can significantly impact the quality of life (QoL) and compliance to anticancer treatments through several mechanisms (Figure 1); in fact, cancer patients, due to their compromised immune systems, may be at higher risk of developing PACS compared to the general population [7]; additionally, chemotherapy or immunotherapy regimens can further weaken their immune response and increase their susceptibility to prolonged COVID-19 symptoms [8,9]. Obesity, especially visceral obesity, has been identified as a shared risk factor for cancer and severe COVID-19 or PACS and may also impact the recovery process through several mechanisms (Figure 1) [10,11].

Among different risk factors affecting cardiovascular diseases (CVDs) in cancer patients, a strict interaction between adipocytes, especially white adipose cells (WAT) surrounded by immune cells, can interact with cancer cells [12], inducing adipokine dysfunctions [13], high levels of leptin [14], low levels of adiponectin [15] and interleukin-10 (IL-10) [16], insulin resistance and metabolic syndrome [17]. Notably, cancer-induced hyperuricemia, hyperkaliemia, acute kidney injury (AKI), anorexia, sarcopenia, and high systemic levels of TNF-α, IL-1β, IL-6, and IL-18 can impair heart function, induce myocardial fibrosis and cardiomyocyte atrophy [18,19,20]. As described in the literature, obesity can contribute to persistent symptoms of PACS and could increase the risk of cardiovascular complications [21]; therefore, especially in patients particularly vulnerable, including those with cancer and/or cardiovascular diseases, appropriate weight management strategies, including a balanced diet at low glycemic index and low inflammatory index, regular exercise, should be implemented under the guidance of cardiologists, nutritionists, and oncologists to address weight-related concerns [22,23]. This narrative review will analyze the close interaction between PACS and obesity and cardiovascular diseases in cancer patients, highlighting the molecular pathways involved. The guidance and recommendations provided here should serve as general considerations that could be useful for optimized individualized management of PACS symptoms in cancer patients with/without cardiovascular diseases in order to improve oncological and cardiovascular prognosis.

## 2. Methods

A systematic search of the Medline and EMBASE databases was performed to identify all potentially relevant English-language scientific papers containing original articles on post-acute COVID-19 syndrome in cancer patients. The following search strings were used in PubMed: “Long COVID-19 and cardioncology” OR “Long COVID-19 OR Post acute COVID-19 syndrome AND cancer patients” OR “Post Acute COVID-19 OR Long COVID-19 AND cardiovascular”. The same research criteria were applied to the Clinical Trial Register. The databases were last accessed on 28 May 2024.

## 3. Post-Acute COVID-19 Syndrome: A Clinical Scenario

Long COVID-19 is a condition where patients continue to experience a range of symptoms and health issues weeks or even months after their initial COVID-19 infection has resolved [24]. The clinical characteristics of PACS can vary widely from person to person, but some common symptoms and features are the following (Figure 2): first of all, more than 95% of patients describe a persistent and debilitating fatigue that can be severe and significantly impact a person’s ability to carry out daily activities [25]. Moreover, many long COVID-19 experience shortness of breath and difficulty breathing, even after the initial infection has cleared [26]. Often referred to as “brain fog”, PACS patients may experience difficulties with memory, concentration, and cognitive functions, most probably due to systemic pro-inflammatory cytokines that reduce glutaminergic signaling in neurons [27]. Importantly, some patients with PACS continue to experience chest pain or discomfort, which can be related to inflammation of the heart tissue or lungs, and these symptoms are always associated with persistent muscle and joint pain, similar to flu-like symptoms, that reduce daily physical activity and increasing sarcopenia [28,29]. Other symptoms of PACS-affected patients include frequent episodes of headaches, anosmia (loss of smell) [30] and dysgeusia (altered or loss of taste) [31] that may persist for several months, heart palpitations, chest tightness, or irregular heartbeats, possibly due to inflammation of the heart muscle (myocarditis) [32]. Moreover, frequent digestive issues, including diarrhea, abdominal pain, and nausea, have been reported as part of long COVID-19 symptoms [33]; 10–15% of patients reported skin rashes, hives, or other dermatological symptoms [33]. Another very frequent clinical symptom of PACS includes anxiety and depression, which could possibly be due to the physical and emotional toll of the prolonged illness [34]. It is important to note that the clinical characteristics and severity of long COVID-19 can vary widely, and new symptoms and complications are still being discovered as research continues. Additionally, in the clinical scenario of PACS-affected patients, significant changes in intestinal microbiota are described [35], exposing them to mucositis and reduced micronutrient adsorption [36]. Moreover, autoimmune diseases were associated with PACS, including thyroiditis, Graves’ disease, and autoimmune arthritis [37]. Based on this clinical picture, proper management of long COVID-19 may involve a multidisciplinary approach, including medical, psychological, and nutritional therapies to address the range of symptoms and their impact on quality of Life (QoL) and cardiovascular disease risk.

## 4. Post-Acute COVID-19 Syndrome and Cardiovascular Complications in Cancer Patients

Cancer patients are strictly exposed to high cardiovascular complications, also due to PACS [38]; especially lung cancer patients, characterized by high rates of morbidity and mortality, could be exposed to significant symptoms [38]. Patients with cancer under therapy with chemo-radiotherapy regimens are also more sensitive to SARS-CoV-2 infection [39]. According to recent data, the death rate of cancer patients with COVID-19 is 22.4%; however, for thoracic malignancies, this number rose to 33% [40]. Furthermore, cancer patients with COVID-19, especially those with lung cancer, may be more vulnerable to long-term issues and the emergence of post-acute syndrome [41]. A recent review focalized on the need to expand the knowledge about interventions to manage PACS syndrome in cancer patients through a multidisciplinary approach [42,43,44]. Early research described the acute and subacute effects of COVID-19 on different organs, including the lungs and heart; however, more recent research has focused on the chronic symptoms associated with COVID-19 [45,46,47,48]. There is little information on the prevalence of PACS in cancer patients and how it influences their condition, care, and therapy [49]. Recent reports reported a frequency of PACS in the general population range greatly from 10% to 87% [50,51]; however, cancer patients with PACS experienced fatigue (82%), sleep disturbance (78%), myalgias (67%), gastrointestinal symptoms (62%), headache (47%), altered smell and taste (47%), dyspnea (47%), and cough (46%) [50,51].

In brief, the mechanisms of PACS-associated symptoms involve an immune-related response to SARS-CoV-2 S protein; in fact, patients with extended COVID-19 symptoms showed an enhanced antigen-specific CD4+ T cell response, together with a longer T cell response magnitude and an increased expression of PD-1-expressing T lymphocytes, which suggested a T cell exhaustion [52,53]. Queiroz et al. found that patients with PACS had greater concentrations of the pro-inflammatory cytokines (IL-17 and IL-2) compared to control groups, indicating a Th-1-derived pro-inflammatory systemic condition [54]. These data associated PACS with chronic inflammation in the injured organs [55,56]. High levels of pro-inflammatory cytokines are associated with cancer growth and chemotherapy-related cardiovascular side effects [57]. Oncogenesis appears to be largely influenced by several cytokines, including interleukin-1 (IL-1) and interleukin-6 (IL-6) [57]. Furthermore, cancer-associated symptoms involve TNF-alpha and IL-1β, which are key orchestrators of chemotherapy-related cardiomyopathies (CTRCDs), myocarditis, vasculitis, and cardiac fibrosis [58]. Therefore, cancer patients with PACS could be exposed to additional cytokine storm-related pro-inflammatory effects due to the combinatorial persistence of cancer diseases, CTRCDs, and immune reaction against COVID-19 [59,60,61]. Interestingly, the antiviral treatment with remdesivir in hospitalized COVID-19 patients showed a protective impact on long-term COVID-19 symptoms, according to a recent prospective study (*p* < 0.001) [62]. These observations may be the result of a reduction in chronic inflammation due to a reduced virus replication period. Additionally, nirma-trelvir/ritonavir demonstrated a significant reduction in viral levels and systemic chemokines and cytokines in a non-randomized, controlled experiment conducted in Shanghai [63]. In patients with active cancer and under chemo-radiotherapy regimens, the female sex and the diagnosis of diabetes mellitus appear to be strongly associated with prolonged COVID-19 symptoms [64,65,66,67]. Interestingly, high glucose levels increase SARS-CoV-2 intake in human cells and exacerbate virus-related damages in several tissues due to stimulation of glycation [65]; advanced glycation end products (AGEs) stimulated Galectin-3 and AGEs receptors that increased ERK and NF-kB signaling in human cells predisposing the patient to very high levels of inflammatory cytokines and chemokines, which increase cardiovascular risk, loss of muscle mass (sarcopenia) and sarcopenic obesity, i.e., the reduction in muscle mass associated with high levels of visceral fat [66,67], as described in details in paragraph 5.0.

## 5. Post-Acute COVID-19 and Visceral Obesity

Notably, all adipocytes and adipose-like cells act as a reservoir for SARS-CoV-2 during acute COVID-19 infection and act as key inductors of systemic inflammation [68,69,70]. Figure 3 illustrates the key role of brown adipose tissue (BAT) reduction in the pathophysiology of PACS [71,72]. White adipose tissue (WAT), very rich in patients with visceral obesity and metabolic syndrome [73], interacts directly and indirectly with SARS-CoV-2, reducing BAT levels and functions. Notably, high levels of WAT were also seen in patients with thin-outside-fat-inside (TOFI) but without a history of metabolic disorders [73,74]. SARS-CoV-2 interacts directly with angiotensin-converting enzyme 2 (ACE2), which is widely expressed on endotheliocytes of stromal vascular tissue, a crucial component of BAT, and adipocytes in WAT, which are primarily found in the visceral tissue. Notably, SARS-CoV-2 interaction with visceral obesity increases VEGF, free fatty acids, cytokines, and other pro-inflammatory chemokines that contribute to the systemic inflammation seen in PACS patients: the name of the overall process is a vasculature–adipocyte-induced systemic metaflammation [75,76,77,78,79,80]. Furthermore, endothelial dysfunction, accelerated coagulation, long-term changes in hemostasis, and chronic inflammation are all caused by vasculature–adipocyte-induced metaflammation and associated mechanisms related to nitric oxide and hydrogen sulfide interaction [81,82,83,84]. These factors, along with vascular permeability factor/vascular endothelial growth factor (VPF/VEGF), may significantly raise the risk of PACS [85]. More in detail, patients with PACS, due to sedentary and systemic low-grade inflammation, have a reduced content of brown adipose tissue, characteristic of lean patients, that is rich in adipocytes with M2-polarized macrophages and high levels of Th2 lymphocytes and Treg cells that induces an anti-inflammatory microenvironment in BAT [84]. In brief, PACS-affected patients are characterized by high levels of WAT with a switch Th1/Th17 microenvironment, such as adipocytes surrounded by M1 macrophages and Th1 and Th17 lymphocytes [83]. WAT with a Th1/Th17 polarization induces adipokine dysfunctions, high levels of leptin, insulin resistance, metabolic syndrome, and low levels of adiponectin and resistin.

The findings of a recent study summarized that SARS-CoV-2 interaction with WAT increases the severity of COVID-19 through two mechanisms that could operate in a feed-forward loop [86]: (i) viral amplification within adipocytes [86] and (ii) local and systemic inflammation, which is primarily caused by inflammatory responses from macrophages that reside in adipose tissue and are infected abortively [87,88]. In fact, several studies focus on preventing strategies of adipocyte and macrophage M-1 accumulation and interaction in PACS, thereby preventing inflammatory responses in adipose tissue [89]. Treatment of metabolic disorders associated with obesity involves an anti-inflammatory diet based on foods with low glycemic and insulinemic index [90], and it could be very useful in PACS-affected patients. Moreover, cyclooxygenase inhibitors have been shown to have anti-inflammatory and antiviral activity against both DNA and RNA-based viruses and exert beneficial effects in COVID-19 patients due to their reduction in adipose tissue inflammation in obese patients [91]. Furthermore, considering that inflamed adipocytes serve as a reservoir for viral infection, as a continuous source of replication, anti-inflammatory drugs or low glycemic diet may contribute to reducing a prolonged COVID-19 syndrome in obese patients, and this approach needs to be investigated in further research [92].

Special attention should be paid to cancer patients with high levels of visceral obesity [93]. Visceral obesity refers to the accumulation of fat in the abdominal cavity, specifically around vital organs such as the liver, pancreas, and intestines [94]. Visceral fat is associated with a higher risk of many diseases, including heart failure, atherosclerosis, diabetes, cancer, and metabolic syndrome [95]. The potential effects of long COVID-19 on visceral obesity are not yet fully understood, but several factors could contribute to an increased risk of visceral obesity in patients with long COVID-19, such as physical inactivity, changes in appetite, metabolic changes, and high stress [96,97,98]. In detail, many patients with PACS experience persistent fatigue and reduced exercise tolerance, which can lead to a more sedentary lifestyle [99] that increases visceral obesity. Some PACS patients experience changes in appetite and taste, which may result in altered dietary habits, including more frequent consumption of salt and high-fat foods that are key orchestrators of visceral obesity [100]. Moreover, PACS can lead to insulin resistance and glucose metabolism dysfunctions; these changes may promote fat storage, including visceral fat [101]. High levels of insulin seen in PACS patients play a key role in the incidence and recurrence of cardiovascular diseases and cancer through several mechanisms, including overstimulation of LKB1/mTORC1 pathways [102].

An uncommon visceral fat depot is the epicardial adipose tissue (EAT) that is anatomically and functionally near the myocardium and coronary arteries [103]. It has been previously shown that EAT may act as a functional reservoir and amplifier for SARS-CoV-2 [104,105]. Interestingly, due to its close proximity to the heart and elevated levels of inflammatory secretions, EAT has been proposed as a potential contributor to the pathogenesis of myocarditis associated with COVID-19 and PACS [105]. Human EAT133 expresses angiotensin-converting enzyme 2 (ACE2), which is well known to be the receptor that allows SARS-CoV-2 to intake cardiomyocytes [106]. In a mouse model, angiotensin therapy also decreased inflammatory cytokines in EAT, but downregulation of ACE2 levels increased EAT inflammation [106]. Thus, the regulation of ACE in EAT may contribute to the cardiac and perivascular inflammation associated with COVID-19 and PACS. ACE inhibitors may be used in treatment for certain COVID-19 aftereffects, albeit there is currently insufficient evidence of this [107]. Notably, EAT of hospitalized patients with severe or critical COVID-19 shows signs of increased inflammation on computerized tomography (CT) scan; in patients with COVID-19, EAT density on CT is markedly elevated at hospital admission and decreases to normal at discharge, whereas subcutaneous fat shows no signs of inflammation [108]. EAT inflammation decreased in patients with COVID-19 or PACS patients who received oral or intravenous dexamethasone, whereas no significant changes in inflammation were observed with other COVID-19 therapies. Therefore, EAT might have a role in COVID-19-related cardiac syndrome, and CT-measured EAT attenuation could be a marker of inflammation and severity of COVID-19 and PACS also [109].

## 6. Post-Acute COVID-19 and Inflammation

Not-well-known players of cardiovascular mortality risk in cancer and non-cancer patients with PACS are sarcopenic obesity and myosteatosis [110]. In detail, patients with PACS are exposed to high systemic levels of sarcopenic cytokines, namely IL-6 and TNFα, which reduce the number and functionality of mitochondria in skeletal muscle cells, inducing sarcopenia [111]. On the other hand, PACS results in high levels of WAT surrounded by M1 macrophages, Th-1 and Th-17 lymphocytes [112] that induce a pro-inflammatory adipose microenvironment that reduces systemic levels of IL-10 (with anti-inflammatory action), increases systemic levels of leptin, irisin, insulin, and IGF-1, downregulating GLUT-4 expression on muscle cells [113]. The combination of these factors induces sarcopenic obesity. Furthermore, PACS patients with high levels of WAT have high concentrations of systemic free fatty acid (FFA) that accumulate between muscle fibers, called intramuscular triglycerides (IMTGs), leading to myosteatosis [114]. IMTGs are extremely dangerous for cancer and non-cancer patients [114]; biochemically, IMTGs increased Plin-2 levels, a small molecule with epigenetic properties able to reduce muscle mass and strength, thus inducing mitochondrial dysfunction and autophagy in the muscle and resistance to anabolism [115]. Clinically, sarcopenia and myosteatosis have been identified as new prognostic markers of cardiovascular mortality, cancer incidence and recurrence, and atherosclerosis [116]. Myosteatosis was associated with age, sarcopenia, hypertension, type 2 diabetes, chronic obstructive pulmonary disease, cancer, heart failure, and ischemic stroke [117]. In univariable analysis, reduced survival was linked with both sarcopenia (hazard ratio [HR], 2.82; 95% confidence interval [CI], 2.05–3.86; *p* < 0.001) and myosteatosis (HR, 4.13; 95% CI, 3.03–5.63; *p* < 0.001). Myosteatosis (HR, 2.09; 95% CI, 1.46–2.99; *p* < 0.001) and sarcopenia (HR, 1.40; 95% CI, 0.97–2.01; *p* = 0.073) were not related to worse overall survival in the multivariable model after adjusting for other prognostic indicators [118]. Compared to the general population, patients with myosteatosis have a significantly increased risk of cardiovascular death, arrhythmia, myocardial infarction, and hypertension. A high risk of type 2 diabetes, hospitalization, endothelial dysfunction, cancer, and hypercholesterolemia are strictly linked to myosteatosis [119]. Moreover, myosteatosis was associated with lower overall survival (OS) in patients with lymphomas, gynecological, renal, periampullary/pancreatic, hepatocellular, gastric, and colorectal cancer [120,121]. Furthermore, myosteatosis during the postoperative recovery had a detrimental effect on the non-indication of adjuvant therapy in colorectal cancer patients [121]. According to several studies, patients with both sarcopenia and myosteatosis had the greatest 7-year mortality rate (94.45%), whereas those who had neither ailment had the lowest mortality rate (83.31%) [122]. Myosteatosis was found to be substantially linked to a poor transarterial chemoembolization response and decreased survival [122]. Prior to transarterial chemoembolization, identifying patients with myosteatosis may enable early therapies to preserve muscle quality and may enhance prognosis in patients with hepatocellular carcinoma [122]. It is interesting to note that a recent review of the literature reveals that low muscle density is a predictor of OS for women with metastatic breast cancer and that patients with sarcopenia have more severe toxicity related to chemotherapy, shorter OS, and increased tumor progression compared to patients with normal skeletal muscle composition [123]. In line with the literature, body composition assessment is useful as a breast cancer predictive metric in clinical practice. A recent multicenter analysis looked into the prognostic significance of body composition in 447 COVID-19 patients [124]. In brief, myosteatosis was found to be independently correlated with 30-day mortality (odds ratio = 2.72; 95% confidence interval [CI], 1.71–4.32); *p* < 0.0001). In COVID-19 patients, myosteatosis, as determined by chest CT, is linked to 30-day mortality; in conclusion, the author concluded that patients with myosteatosis could be defined as a high-risk group [124]. In a different study, regardless of age, sex, or body mass, myosteatosis seen on acute COVID-19 substantially predicts persistent dyspnea and mobility issues six months following hospital discharge [125]. In patients with sarcopenia and myosteatosis, insulin receptor sensitivity is highly dysregulated, resulting in insulin resistance. Insulin resistance in PACS patients could result from chronic systemic inflammation [126]. Through established mechanisms, high cortisol levels can exacerbate insulin resistance in PACS patients [127]. Furthermore, corticosteroids used in COVID-19 and PACS patients exacerbate insulin resistance through endogenous cortisol-related pathways [128]. Another clinical aspect able to exacerbate insulin resistance is reduced physical activity in PACS patients, resulting in sarcopenia and myosteatosis, through positive feedback mechanisms of insulin resistance [129]. In brief, the WAT increase and myosteatosis associated with sarcopenia reduce insulin receptor functions and increase glucotoxicity and cardiovascular mortality [130]. Moreover, recently, some direct viral effects on insulin synthesis were demonstrated. The SARS-CoV-2 virus may have direct effects on insulin-producing cells in the pancreas, potentially worsening insulin resistance [131]. High systemic levels of C-reactive protein (CRP), cytokines, chemokines, and growth factors are induced by insulin resistance [132]. C-reactive protein is a liver-derived marker of systemic inflammation due to tissue damage, infection, and PACS. Significant increases in CRP can be seen in acute COVID-19 patients [133]; however, PACS patients experience high CRP levels for several months after the first contact with SARS-CoV-2 [134], increasing the risk of multi-organ damage [135]. Moreover, COVID-19 in the acute phase and PACS are associated with high levels of interleukin-6 [136]. Interleukin-6 (IL-6) is a cytokine secreted by immune cells, cardiomyocytes, and cancer cells [137]; it is essential for controlling the body’s immunological response to inflammation and infections [138]. According to recent work, fatigue, cognitive fog, joint pain, and systemic symptoms seen in PACS patients are significantly associated with IL-6-related overexpression [139] (Figure 4). In fact, the treatment strategies for acute COVID-19 and PACS could involve IL-6/IL-6 receptor-blocking agents, such as sarilumab or tocilizumab, characterized by significant beneficial properties [140].

As previously described, patients with PACS have high WAT levels [141]; as summarized in Figure 3 and Figure 4, WAT interacts with several immune cells, including M1-polarized macrophages, which decreases IL-10 secretion and adiponectin production by adipocytes [142], resulting in a pro-inflammatory microenvironment. Adipose tissue is able to produce adiponectin, which is involved in the reduction of inflammation and the improvement of insulin sensitivity [143]. In addition to its anti-inflammatory qualities, adiponectin has also been linked to immune response modulation during COVID-19 and PACS, according to some research findings [144]. Patients with obesity, metabolic syndrome, cancer, and cardiovascular illnesses frequently have low levels of adiponectin [145]. Recent advances in clinical research in cardio-oncology suggest exploring whether interventions to modulate adiponectin levels, such as lifestyle changes, anti-inflammatory died, and nutraceuticals, could potentially influence the course PACS and associated cardiometabolic risk. CXCL-12 (stromal cell-derived factor 1, SDF-1) is another cytokine associated with PACS that is involved in tissue healing, immunological cell trafficking, cancer, and heart failure [146]. It is essential for controlling the migration and targeting of immune cells and stem cells to several tissues, including the lungs and heart, and could play a key role in COVID-19 pathogenesis [147]. More in detail, SARS-CoV-2 targets lungs, heart, and adipocytes, stimulating the recruitment of immune cells to these organs through the overexpression of CXCL-12 and other pro-inflammatory cytokines [148,149]. Patients with metabolic syndrome and visceral obesity have high FFA levels, able to reduce mitochondrial metabolism, inducing heart failure (HF) in patients with or without cancer and PACS [150]. A pharmacological strategy used to restrict de novo lipogenesis is based on the blocking of fatty acid synthase or ATP citrate lyase, which catalyzes the conversion of citrate obtained from glucose to acetyl-CoA [151]. Fatty acid synthase inhibitor-targeted clinical trials have demonstrated efficacy in treating metastatic KRAS mutant non-small cell lung cancers and metastatic HER2 breast cancer (currently undergoing phase II clinical trials) [152]. Nevertheless, a comparable association has not been observed in trials for heart failure. Alternative promising methods are based on the activation of peroxisome proliferator-activated receptor α or on the selective inhibition of carnitine palmitoyl-transferase 1 and 3-ketoacyl coenzyme-A thiolase [153]. Notably, these therapies reduced circulating FFA levels and reduced cardiovascular events in cancer patients, but no studies have described the outcomes in PACS patients to date. However, a mitochondrial-targeted drug, metformin, was studied in COVID-19 and PACS patients [154]. Metformin lowers insulin and insulin-like growth factor (IGF) 1 plasma levels as well as mitochondrial ATP production [155]. These effects underscore the anticancer characteristics of metformin, especially in tumors with high levels of organic cation transporters, by increasing reliance on glycolysis for ATP provision and making cancer cells more susceptible to restricted glucose availability [156,157,158]. Notably, metformin use is able to reduce overall mortality in COVID-19 patients and to prevent PACS through the reduction of glucotoxicity, inflammation, visceral obesity, and free fatty acid levels [159]. Furthermore, it has been proposed that SARS-CoV-2 stimulates the activation of palmitic acid production by upregulating the transcription-signaling genes of fatty acid synthase (FASN), acetyl-CoA carboxylase (ACC), and stearoyl-CoA desaturase 1 (SCD1) [160]. Through this process, the virus raises the body’s supply of lipids and encourages more viral reproduction, which raises the viral load [161]. Behenic and lignoceric acid levels, two saturated fatty acids, changed significantly in patients with PACS compared to control [162]. In one other study [163], behenic acid deficiency was seen in COVID-19 patients and associated with unfavorable clinical outcomes, such as changed blood metabolites and intestinal inflammation [163]. While lignoceric acid deficiency has been associated with adverse immunological responses, especially in autoimmune illness and cancer treatment response, levels of behenic acid in COVID-19 patients have not been investigated before [164]. Higher levels of lignoceric acid have also been associated in other studies with a lower incidence of age-related diseases, suggesting that the acid may have a protective function in the body [165]. These preliminary results showed different plasma fatty acid profiles in PACS and acute COVID-19 patients, stimulating further investigation into the role of plasma fatty acid in the pathogenesis of PACS [166]. Dietary supplementation of behenic, lignoceric, linoleic, GLA, and EPA may be an economical and non-invasive way of preventing or controlling PACS, especially in overweight or obese people [167]. As described in paragraph 1, given the frequent involvement of multiple organs, COVID-19 consequences may ideally necessitate a multidisciplinary strategy [168]. Among them, exercise has a compelling case for its application in patients who are particularly vulnerable, such as cancer patients [169]. For example, metastatic lung cancer patients with persistent symptoms, including fatigue, physical deconditioning, headaches, and peripheral neuropathy, have been reported during PACS [169]; in these patients, physical exercises associated with nutraceutical administration and proper anti-inflammatory nutrition were able to improve PACS-related clinical scenario and to reduce comorbidities [170,171].

## 7. Suggestions to Reduce Cardiovascular Complications in Cancer Patients with Post-Acute COVID-19 Syndrome

As summarized in Figure 5 and Table 1, cardiologists and oncologists should include pharmacological and non-pharmacological strategies to reduce metabolic risk factors in cancer patients with PACS [172]. In line with AHA and European Association for Cancer Prevention (WCRF) guidelines, suggestions on a proper anti-inflammatory diet and daily passive or active physical activity should be promoted for primary and secondary prevention of cardiomyopathies [171]. Supplements based on arginine associated with ascorbic acid, N-acetyl cysteine (NAC) alone or associated with multivitamins, tocotrienols, and taurin have been shown in recent randomized clinical trials to significantly improve lung functions, mental health, fatigue, and cardiovascular parameters in less than six months in non-cancer patients with PACS [173,174]. A dietary supplement based on NAC and vitamins avoided PACS-related tiredness in 28 days and enhanced exercise tolerance, according to a Gemelli Hospital study [175]. Another study found that nutritional supplements containing ascorbic acid, NAC, zinc, and iron (known as “Apportal”) significantly improved QoL and reduced fatigue in PACS patients as compared to individuals who were not receiving treatment [176]. Oral supplement with L-arginine is often recommended to enhance muscular growth and resistance in elderly patients as well as agonist sports [177]; in summary, it maximizes ATP synthesis to enhance mitochondrial functions. L-arginine combined with ascorbic acid supplementation enhanced walking ability, muscle strength, endothelial function, and fatigue in patients with prolonged COVID-19 infection in a recent clinical trial [178]. It is well known that oral administration of L-arginine in patients with SARS-CoV-2 infection reduces systemic levels of pro-inflammatory and atherogenic cytokines, such as IL-6, IL-1, and IL-2 [179]. We also hypothesize that this medication may be helpful to long-term COVID-19 patients. According to a recent analysis, vitamin D insufficiency is linked to delayed recovery from PACS, whereas zinc deficiency is linked to both acute and chronic inflammation [180].

The recruitment of neutrophil granulocytes requires zinc [181]. Along with activating CD4+ and CD8+ T cells, it plays a crucial role in phagocytosis, oxidative burst production, and chemotaxis [182]. Significantly, zinc shortage in lung cells, alveolar macrophages, and hepatic macrophages in RES can impair lung barrier function, resulting in cardiometabolic disorders and Acute Respiratory Distress Syndrome. In COVID-19 patients, vitamin D insufficiency is associated with both disease severity and mortality [183]. Acute Respiratory Distress Syndrome, obesity, and cardiovascular mortality were seen in COVID-19 patients with vitamin D insufficiency [184]. Interestingly, another nutraceutical strategy of great interest in PACS control is the liposomal ascorbic acid; it can remain in the plasma for a few minutes after delivery and has a high intestinal absorption rate, which promotes pro-immune and antioxidant effects in PACS patients [185]. After receiving liposomal ascorbic acid twice a day, patients with sarcopenia and fatigue showed considerable improvements in their muscle metabolism. Sarcopenia and myosteatosis are frequently linked in patients with cancer and cardiovascular illnesses, which raises their death rate [186]. PACS exacerbates skeletal muscle loss by reducing satellite muscle cells and mitochondrial content in muscle cells, which are caused by sarcopenic cytokines (IL-6 and TNF-α) [187]. Multiple amino acid administration orally (L-Leucine, L-Valine, L-Isoleucine, L-Lysine hydrochloride, L-Phenylalanine, L-Threonine, L-Methionine, and L-Tryptophan) may be beneficial in the prevention and treatment of sarcopenia, enhancing muscle anabolism and physical resistance [188]. Multifunctional glycoprotein lactoferrin, which is present in milk and other body secretions, has drawn interest due to its possible application in the treatment of both viral and non-viral illnesses, such as myocarditis, vasculitis, cancer, and COVID-19, which includes PACS-related symptoms [189]. Lactoferrin modulates the immune response to demonstrate anti-inflammatory properties. Lactoferrin’s ability to reduce inflammation may help alleviate some symptoms in PACS, where inflammation may continue; moreover, recent clinical studies evidenced direct antiviral effects of lactoferrin against SARS-CoV-2 [190]. Lactoferrin may cause inflammatory responses in PACS patients, alleviating clinical symptoms. By strengthening innate immune responses and encouraging a balanced immunological response, lactoferrin has the ability to modify the immune system [191]. Lactoferrin’s immunomodulatory actions may be helpful in PACS, as immune system dysregulation may be a factor in chronic symptoms. Lactoferrin has been shown to have the ability to stimulate the growth of beneficial bacteria in the small intestine [192]. Given that PACS has been linked to gut dysbiosis, lactoferrin’s impacts on gut health may have an indirect effect on symptom management. Moreover, lactoferrin administration is able to reduce systemic levels of IL-6 in COVID-19 and PACS patients [193]; ferritin and D-dimer serum levels have also been reported to have decreased. Another nutraceutical, called bromelain, has recently shown beneficial properties in PACS [194]. Anti-inflammatory and fibrinolytic properties have been investigated during bromelain therapy [195]. Although there is not much evidence specifically linking bromelain to PACS management, its anti-inflammatory qualities might be useful in treating some of the condition’s symptoms [196]. Through the inhibition of specific pro-inflammatory cytokines and mediators (IL-1, IL-6, and IL-17), bromelain has been demonstrated to modify inflammatory pathways and reduce inflammation [197]. The anti-inflammatory qualities of bromelain may help reduce some of the symptoms associated with PACS, where inflammation may linger and contribute to symptoms [198]. According to recent findings, bromelain has mucolytic properties, which means it can aid in the breakdown of mucus [199]. People with respiratory symptoms, which are frequent with PACS and include continuous coughing or congestion, may benefit from this nutraceutical. Furthermore, by affecting the activity of immune cells (NK cells, CD3+ and CD8+ cells), as well as cytokines, bromelain may influence immunological responses [200]. Notably, patients with PACS treated with bromelain showed improved NK cell activity [201]; more specifically, as demonstrated by Kritis et al. [202], bromelain can inhibit SARS-CoV-2 intake in human cells by cleaving its spike protein and decreasing the expression of ACE2 and Transmembrane protease, serine 2 (TMPRSS2), an enzyme related to estrogens, testosterone, and insulin levels [202]. Moreover, TMPRSS2 hydrolyzes glycosidic residues in human cell membranes that are very useful to NK and APC cells and CD-8+ lymphocytes to interact with cancer cells [203]. Bromelain reduces inflammatory mediators and NF-kB, which, in turn, downregulates the pro-inflammatory prostaglandin E-2 (PGE-2) to reduce inflammation [204]. Bromelain has been used at a daily dosage of 200–2000 mg; however, the recommended beginning dose in clinical trials is 500 mg [205]. However, Bromelain can increase bleeding risk; therefore, even though it has a low toxicity level and is typically safe, it should be administered cautiously to patients who have cardiovascular disease or cancer [206].

Patients with COVID-19 have multiple symptoms, including those related to the respiratory and gastrointestinal systems [207]; therefore, reliable data supporting the efficacy of probiotics in alleviating these symptoms are lacking. A recent meta-analysis concluded that probiotics may be able to reduce COVID-19-related inflammatory response, overall symptoms, and length of hospital stay [208]. By improving the intestinal flora and shortening the length of diarrhea, probiotics may alleviate gastrointestinal symptoms; through the gut–lung axis, they may also alleviate respiratory symptoms [209]. The most studied probiotics involve Streptococcus thermophilus, Bifidobacterium breve, Bifidobacterium longum, Bifidobacterium infantis, Lactobacillus acidophilus, Lactobacillus plantarum, Lactobacillus paracasei, Lactobacillus delbrueckii and lactobacillus reuteri [210]. By enhancing IgA responses, probiotics like Limosilactobacillus reuteri supplementation associated with vitamin D may increase the long-term effectiveness of mRNA-based COVID-19 vaccinations and could be very useful in PACS management [211]. Cancer survivors should strictly follow the lifestyle and dietary recommendations of the American Institute for Cancer Research (AICR) and the World Cancer Research Fund (WCRF) [212]. Recent clinical research indicates that a low glycemic index diet could improve immune functions and reduce WAT, preserving sarcopenia and myosteatosis through NLRP-3 and MyD-88 signaling pathways [213]. More in detail, an anti-inflammatory diet, rich in polyunsaturated fatty acids and without trans-unsaturated fatty acids, is very useful in reducing ARDS and COVID-19-associated cardiovascular and endothelial diseases [214]. An anti-inflammatory diet is mainly based on plant-based foods, rich in vegetables, fruits, and dried fruit, including almonds, walnuts, hazelnuts, and pine nuts [215]. These foods reduce systemic levels of IL-1, IL-6, and CRP and improve cardiometabolic functions. Moreover, a low glycemic index (GI) diet is also associated with several benefits in patients with PACS, cancer, and cardiovascular diseases [216]. A low GI diet is rich in whole grain cereals and wholemeal bread and requires that, at every meal, there is always a portion of cooked or raw vegetables [217]. Foods avoided should involve white rice, white bread, and refined foods for breakfast, such as over-the-counter croissants, croissants, and rusks, all characterized by a high glycemic index and trans-unsaturated fatty acids [218]. Additional foods to avoid include sugary drinks and all foods rich in glucose syrup, glucose-fructose syrup, and added fructose, characterized by obesogenic and pro-inflammatory activity [219]. Moreover, preserved meats should be avoided, as the European code against cancer says, due to the high quantity of salt and nitrite compounds that induce inflammation of the gastrointestinal mucosa [220]. The overall picture of the review highlights different pharmacological and non-pharmacological approaches to the prevention and management of PACS in particularly vulnerable patients such as those with cancer and cardiovascular diseases. As described in the introduction, cancer survivors are exposed to high risk of cardiovascular and cardiometabolic diseases, including HF, hypertension, T2DM, sarcopenic obesity, arrhythmias, and venous thromboembolism. By analyzing the many clinical pieces of evidence available and the pathophysiological studies on PACS and cardio-oncology, we believe, to the best of our knowledge, that the best preventive and therapeutic approach is based on following an anti-inflammatory, low glycemic, and insulinemic index diet daily, associated with supplementation with arginine, magnesium, and NAC. If the patient has confirmed sarcopenia, oral supplementation with appropriate amino acids must be provided, especially in the early hours of the day, in order to stimulate muscular anabolic processes.
biomedicines-12-01650-t001_Table 1Table 1A list of different therapeutic options of PACS in patients with/without cancer.Therapeutic OptionMolecular PathwayClinical OutcomesRef.Anti-inflammatory diet↓ COX-2, NLRP3, MyD88, pro-inflammatory cytokinesAnti-inflammatory systemic effectsCardioprotective effects[171,172]L-Arginine↑ T-lymphocyte survival, ↑ nitric oxide, cGMP, PKG↓ IL-17, IL-1βPro-immune functions (innate and adaptive immunity)Anti-inflammatory systemic effectsImprovement of endothelial functions[173,174,175]N-acetil cystein (NAC)↑ Cys, GSH, IL-10↓ HOCl, OH-, H_2_O_2_↓ NLRP3, MyD88Anti-inflammatory systemic effectsCardioprotective effects[175,176]Ascorbic acid, NAC, zinc, and iron↑ T-lymphocyte survival↑ nitric oxide, cGMP, PKG↓ IL-17, IL-1β, NLRP3, MyD88Pro-immune functions (innate and adaptive immunity)Anti-inflammatory systemic effectsImprovement of endothelial functions[177,178,179]Cholecalciferol↓ NLRP3, MyD88, IL-6, and IL-17↑ CD8+ T-lymphocyte survival; Natural Killer cells↑ IL-10Anti-inflammatory effectsPro-immune functions (innate and adaptive immunity)[180,181,182,183,184]Liposomal ascorbic acid↓ NLRP3, MyD88, IL-6, IL-17, IL-1β↑ CD8+ T-lymphocyte survival; Natural Killer cells↑ IL-10Anti-inflammatory effectsPro-immune functions (innate and adaptive immunity)[185]Multiple amino acid supplements **(L-Leucine, L-Valine, L-Isoleucine, L-Lysine hydrochloride, L-Phenylalanine, L-Threonine, L-Methionine, L-Tryptophan**)↓ ROS, IL-6, TNF-α, IL-1↑ Mitochondrial biogenesis, motor units, number of fibers↑ Satellite cells function↑ NO, PCG1-αAnti-sarcopenic effectsImprovement of skeletal muscle massImprovement of endothelial functions[187,188]Lactoferrin↓ IL-1, IL-6, and IL-17↑ CD8+ T-lymphocyte survival; Natural Killer cells↑ IL-10↓ ROS, MDA, 4-HNAAntiviral effectsAnti-inflammatory effectsPro-immune functions (innate and adaptive immunity)[189,190,191,192,193]Bromelain↓ IL-1, IL-6, IL-17, PGE2, COX-2↑ CD8+ T-lymphocyte survival; Natural Killer cells↑ IL-10↓ ACE-2, TMPRSS2Antiviral effectsAnti-inflammatory effectsPro-immune functions (innate and adaptive immunity)[194,195,196,197,198,199,200,201,202,203,204,205,206]Probiotics*(Streptococcus thermophilus*, *Bifidobacterium breve*, *Bifidobacterium longum*, *Bifidobacterium infantis*, *Lactobacillus acidophilus*, *Lactobacillus plantarum*, *Lactobacillus paracasei*, *Lactobacillus delbrueckii, and lactobacillus reuteri)*↑ IgA responses↓ IL-17, IL-6, IL-1, TNF-α↑ IL-15, IL-12, IL-21↑ CD8+ T-lymphocyte survival; Natural Killer cellsPro-immune functions (innate and adaptive immunity)Enhancement of antibody production[208,209,210,211]↑: increase; ↓ reduction.

## 8. Conclusions

Post-acute COVID-19 syndrome (PACS) is a chronic condition that increases cardiovascular risk in patients with and without cancer. A deep analysis of PACS pathophysiology and the common pathways with cancer and cardiovascular diseases is strictly necessary. The current review discusses the shared risk factors of cancer, cardiovascular diseases, and long COVID-19-sustained chronic systemic inflammation, as well as the potential preventive strategies, including an anti-inflammatory and low glycemic index diet associated with nutraceuticals, aimed to reduce the magnitude of cardiovascular diseases and overall mortality. Individual follow-up and cardiological management should focus not only on cancer outcomes but also on cardiometabolic risk factors to minimize the risk of serious adverse events in cancer patients with post-acute COVID-19 syndrome.

## Figures and Tables

**Figure 1 biomedicines-12-01650-f001:**
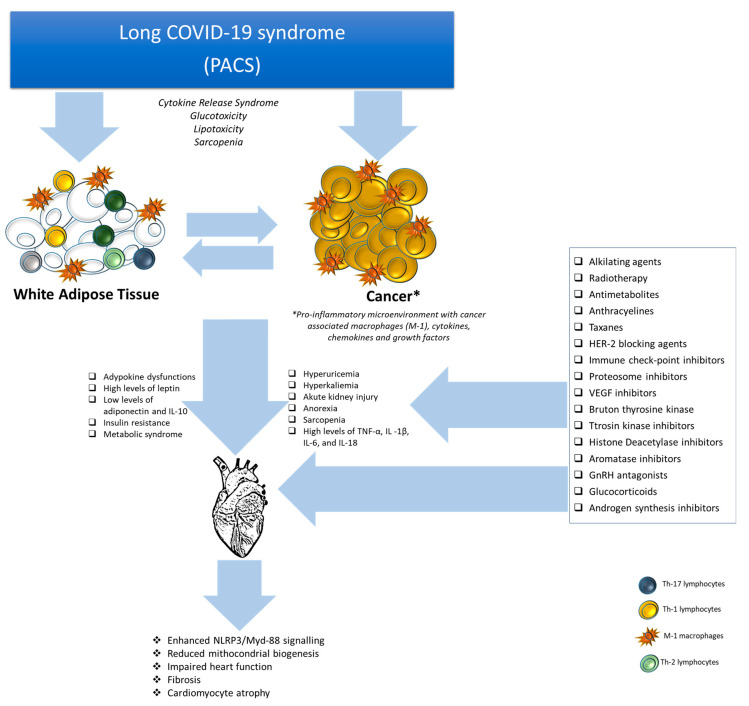
Key roles of white adipose tissue–cancer interaction in post-acute COVID-19 (PACS) syndrome-affected cancer patients and their influence on cardiovascular diseases.

**Figure 2 biomedicines-12-01650-f002:**
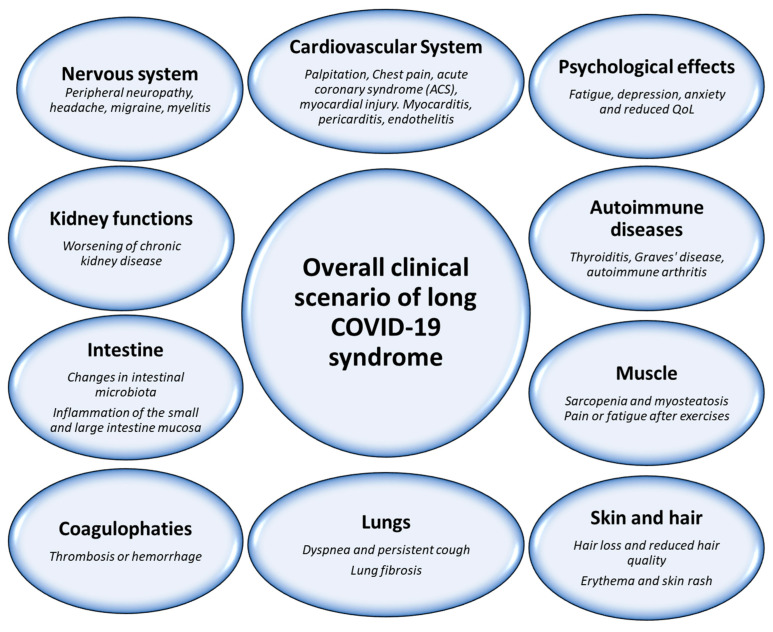
Overall description of post-acute COVID-19 clinical scenario in patients with/without cancer.

**Figure 3 biomedicines-12-01650-f003:**
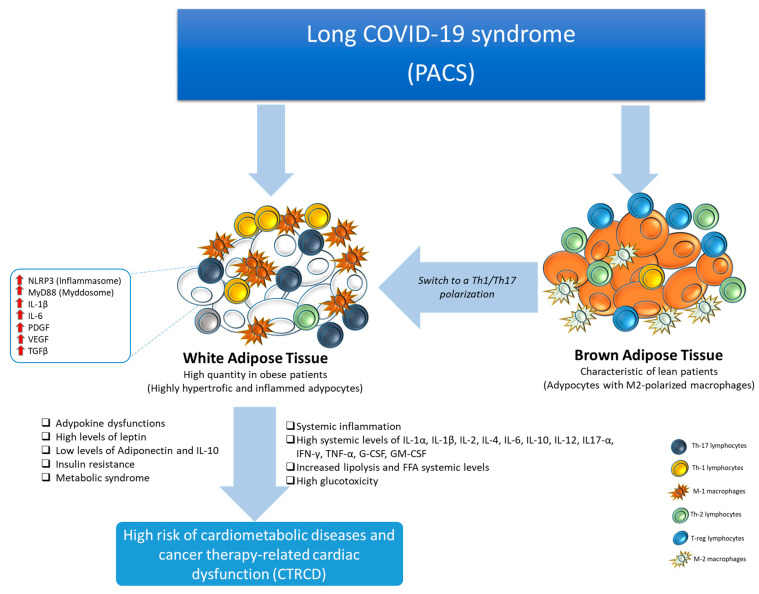
Schematic representation of Th1-Th17 immune polarization in patients with PACS. Brown Adipose Tissue, rich in adipocytes with M2-polarized macrophages and Treg cells, changes their metabolic functions to a Th1/Th17 polarization phenotype in patients with metabolic syndrome alone or associated with PACS. Through direct or indirect pathways, PACS increases white adipose tissue and increases its M1 macrophages and Th1 and Th17 T cells that induce a pro-inflammatory microenvironment. The inflamed white adipose tissue was associated with adipokine dysfunctions involving high systemic levels of leptin and low levels of adiponectin ald IL-10; these patients are exposed to high risk of insulin resistance and metabolic syndrome. Moreover, inflamed white adipose tissue increased lipolysis and FFA systemic levels, as well as several atherogenic cytokines.

**Figure 4 biomedicines-12-01650-f004:**
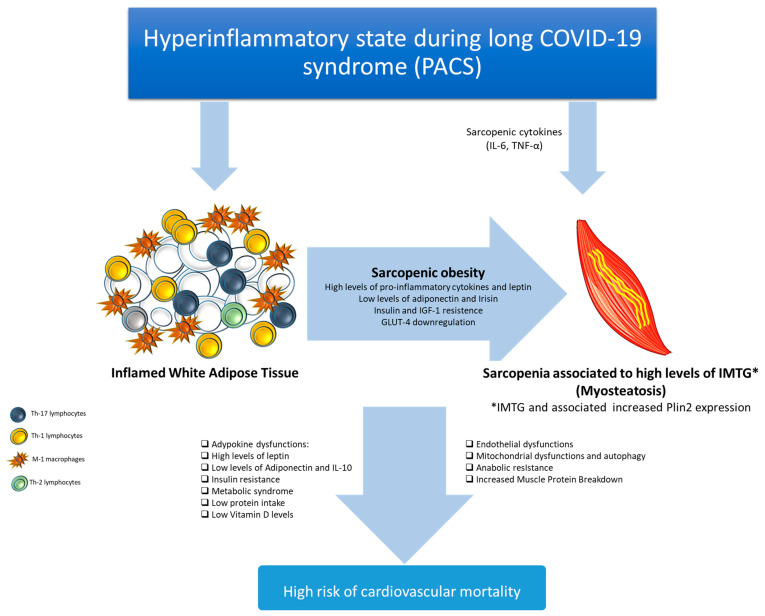
Overall effects of PACS in the incidence of sarcopenic obesity and myosteatosis in patients with and without cancer. PACS-induced hyperinflammatory state is well associated with high levels of circulating cytokines (IL-6 and TNFα), which reduce the number and functionality of mitochondria in skeletal muscle cells, inducing sarcopenia. PACS is associated with high levels of WAT surrounded by M1 macrophages and Th1 and Th17 cells. These lymphocytes induce a pro-inflammatory adipose microenvironment that reduces systemic levels of IL-10 (with anti-inflammatory action) and increases systemic levels of leptin, irisin, insulin, and IGF-1, downregulating GLUT-4 expression in muscle cells. The combination of these factors induces sarcopenic obesity. Furthermore, PACS patients with high levels of WAT have high concentrations of systemic FFA that accumulate between muscle fibers, called intramuscular triglycerides (IMTGs), leading to myosteatosis.

**Figure 5 biomedicines-12-01650-f005:**
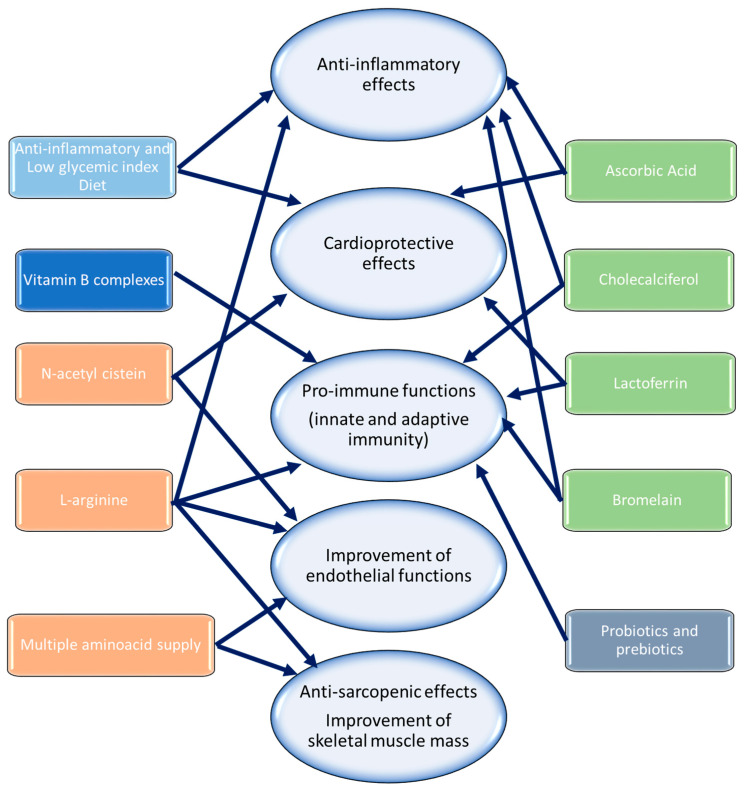
Overview of non-pharmacological therapies of PACS in patients with and without cancer involving L-arginine, N-acetyl cysteine, vitamin B complexes, probiotics and prebiotics, lactoferrin, anti-inflammatory ald low glycemic index diet, bromelin, multiple amino acid supply, cholecalciferol, and ascorbic acid. Different colors correspond to different chemical categories of drugs. The arrows correspond to the most significant clinical effects for each molecule, i.e., anti-inflammatory, cardioprotective, immune-enhancing effects, improvement of endothelial functions, and anti-sarcopenic effects, improving skeletal muscle mass.

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
