# Peer review of "Addressing Post-Acute COVID-19 Syndrome in Cancer Patients, from Visceral Obesity and Myosteatosis to Systemic Inflammation: Implications in Cardio-Onco-Metabolism"

_biomedicines, 2024, doi:10.3390/biomedicines12081650_

Round 1

Reviewer 1 Report

Comments and Suggestions for Authors

.

Author Response

Dear reviewer, we are sorry but you did not provide comments for us to review

Reviewer 2 Report

Comments and Suggestions for Authors

This review focuses on addressing post acute COVID-19 syndrome in cancer patients, from visceral obesity and myosteatosis to systemic inflammation: implications in cardio-onco-metabolism. I think the topic is valuable and interesting. I personally enjoyed this article and think it is a valuable contribution to this subject. There are just a few points of constructive criticism that I think should be addressed:

·         In general, the figures must be edited, the texts are not legible, the font size could be increased. I think the figures can be greatly improved regarding the pathophysiology. The image is also a little hard to appreciate, that quality and resolution should be higher.

·         The numbering of the sections is incorrect, it has to be corrected.

·     In sections 7, tables could be added to summarize the different therapeutic options

·         There were only minor errors in the grammar, but the abbreviations should be checked to be homogenous throughout the article, as well as checking other minor errors such as the justification.

Author Response

We thank reviewer for the useful comments and suggestions aimed to improve the quality of the work and our experience in this interesting field. Here the answers to each point:

  • Ok, we are agree with you. In line with your suggestions, we have modified and improved the quality of the figures, making them more easy to read. Thank you
  • Ok, sorry for the mistake, we have modified this part
  • Ok, thank you for this useful suggestion. We have added a table (table 1)  to summarize the therapeutic options ( see table 1 at pag 14).
  • Ok, thank you, we have added some improvements to make the manuscript more easy to understand (see the underlined parts). Thank you

Reviewer 3 Report

Comments and Suggestions for Authors

The manuscript biomedicines-3069353 "Addressing post acute COVID-19 syndrome in cancer patients, from visceral obesity and myosteatosis to systemic inflamma-tion: implications in cardio-onco-metabolism" aimed to summarize the main metabolic affections of post acute COVID-19 syndrome in cancer patients at low and high risk of cardiomyopathies. 

The purpose of the study is clearly identified and several topics were well discussed, however, in some parts of the manuscript it is necessary include an information about the better strategy against  the post acute COVID-19 syndrome. In another words, it is necessary include the better alternative against the disease, or include which intracelular via could be more important. There are some limitations or weaknesses as described below: 

1) Although the manuscript share and discuss updated information in this topic of study, it is necessary in general aspects insert the position of authors suggesting better strategies or the most important topic in each part of the review. This improvement of the  discussion can improve the manuscript.

2) In the first part of the introduction there some paragraphs with just one sentence, please, improve it.

3) The quality of figure 1 is good, however the painting  of heart (in this figure) is not appropriated. There one square just in this figure. 

4) I suggest that the letters of the words in all figures can not be lower then the letters in the text of the manuscript. It difficult the lecture for readers that print the paper to read it. 

5) in the topic 2 , Methods, there one word with error “CVOVID-19”

6) L121-123 Please , correct the punctuation in this sentence ; eg, “headaches anosmia” without coma.

7) Figure 2 it is very difficult to read because major part of the words are small. There are dozens of different outcomes, however, it is necessary include one title for each circle to address the main information. In my point of view one figure need work as a take home message. In this case, it seems that you just put all outcomes in several balons.

8) line 194 - “paragraph 4.0” This sentence is unclear for me.

9) Figure 3. It is very good figure but take care with low letters. Do you think that is necessary the squares in some words alone, like Th1 macrophages, Th1 cells? May be you can insert different colors in the background inside of the big squares in this figure.

10) In the session 5 are discussed the inflammatory pathways of this disease, however, the figure 4 needs receive the word “inflammatory” highlighted. Besides, the word “inflammatory” should be appears in the legend.

11) In my point of view the paragraph starting in L-395 could be together with the previous one. Both discussed the same topic. 

12) The topic 7, L492 is very good. However, I detected minor errors with paragraphs in L500 and L527 and L557.

13) Figure 5 just insert arrows to the diseases. I really suggest that you can put different colors for some different types of molecules (like peptides , chemicals, ions, … I could choose the better option). Perhaps you can also include some intracelular mechanism to explain each effect.

Comments on the Quality of English Language

The English grammar is very good.

Author Response

We thank reviewer for the useful comments and suggestions aimed to improve the quality of the work and our experience in this interesting field. Here the answers to each point:

  • Ok, we are totally agree with you. We have added in discussion a detailed description of the better pharmacological and non-pharmacological strategies, in our opinion, to reduce cardiovascular events and metabolic diseases in this population cohort ( see pag 16, the underlined parts) . Thank you
  • Ok, sorry for this mistake, thank you
  • Ok, we have modified the Figure 1 according to your suggestions. Thank you
  • We are agree with you, we have modified this part also.
  • Sorry for this mistake, we have corrected this part.
  • Sorry for this mistake, we have corrected this part.
  • Also here, we are agree with you and with your suggestions; based on your suggestions we have modified this figure, making it more easy to read and understand. Thank you
  • Sorry for this mistake, we have corrected this part ( it was: paragraph 5.0)
  • Thank you for this suggestions aimed to improve the quality of the figure. Yes, we have modified this figure to make it more easy to read,
  • Ok, thank you. We have modified this part.
  • We are agree with you! In line with your suggestion, we have modified this part.
  • Sorry for this mistake, we have corrected these sentences.
  • Thank you for these suggestions! Ok, we have modified and improved the quality of the figure 5 in line with your suggestions and added to the revised manuscript file.

Round 2

Reviewer 3 Report

Comments and Suggestions for Authors

In my point of view, the current version of this manuscript is suitable for publication in Biomedicines (ISSN 2227-9059). The subject matter is relevant in this field of knowledge, and the authors have adequately addressed the reviewers' questions. Therefore, I consider this second version of the manuscript appropriate for publication.